# Decreased Survival and Lung Function in Progressive Pulmonary Fibrosis

**DOI:** 10.3390/medicina59020296

**Published:** 2023-02-05

**Authors:** Mark G. J. P. Platenburg, Joanne J. van der Vis, Jan C. Grutters, Coline H. M. van Moorsel

**Affiliations:** 1Interstitial Lung Diseases Center of Excellence, Department of Pulmonology, St Antonius Hospital, 3435 CM Nieuwegein, The Netherlands; 2Department of Clinical Chemistry, St Antonius Hospital, 3435 CM Nieuwegein, The Netherlands; 3Division of Heart and Lungs, University Medical Center, 3508 GA Utrecht, The Netherlands

**Keywords:** PPF, progressive pulmonary fibrosis, survival, ILD, FVC, DLCOc

## Abstract

*Background and Objectives:* Progressive pulmonary fibrosis (PPF) is a recently described term reserved for patients with fibrotic ILD other than idiopathic pulmonary fibrosis (IPF) with fast clinical deterioration. Here, survival and prognostic biomarkers at the time of diagnosis for PPF are investigated in a fibrotic ILD other than IPF cohort (non-IPF). *Materials and Methods*: Patients diagnosed during the period of 2012–2018 at the ILD Center of Excellence (St. Antonius Hospital, Nieuwegein, The Netherlands) with a fibrotic ILD were included in this study. The presence of PPF was investigated using the criteria from the updated IPF/PPF guideline during the first year after diagnosis. Logistic regression analysis was used to determine risk factors for PPF. A Kaplan–Meier survival analysis with log-rank test was conducted to analyze survival in patients with and without PPF. *Results:* This study included 304 non-IPF patients and, for comparison, 379 IPF patients. In non-IPF patients, 146 (46%) fulfilled ≥2 criteria for PPF. These patients had a median transplant-free survival rate of 2.9 ± 0.4 years, which was worse than non-IPF patients without PPF (10.1 ± 1.8 years, *p* < 0.001). The risk for PPF was increased in patients with FVC < 50% (odds ratio (OR) of 2.50, 95% CI = 1.01–6.17, *p* = 0.047) or DLCOc ≤ 35% (OR = 2.57, 95% CI = 1.24–5.35, *p* = 0.011). In the first 3 years after diagnosis, survival in PPF and IPF is the same, while in the following years IPF has a significantly worse survival. *Conclusions:* The non-IPF cohort with PPF had a significantly worse transplant-free survival compared with the non-IPF cohort without PPF. Independent risk factors for PPF in non-IPF were FVC < 50% and DLCOc ≤ 35%.

## 1. Introduction

Interstitial lung disease (ILD) is a heterogeneous group of >200 pulmonary diseases, which are characterized by an inflammatory and/or fibrotic process located at the pulmonary interstitium [1]. Patients may suffer from disabling complaints such as dyspnea and cough, which in some patients may develop into life-threatening respiratory insufficiencies. The clinical course of patients is highly variable as some patients remain stable for a considerable amount of time, whereas others demonstrate fast lung function decline [2].

The hallmark of progressive, fibrotic ILD is idiopathic pulmonary fibrosis (IPF), a disease with an unknown etiology and predominantly diagnosed in older males with a positive smoking history [3,4]. One of the key features in IPF is the radio- and/or histopathological usual interstitial pneumonia (UIP) pattern, which is defined by the presence of honeycombing, traction bronchiectasis and subpleural reticulations with an apicobasal gradient [5]. The survival of patients ranges between 2 and 5 years after diagnosis and is dependent on multiple variables [2,3,6]. One of them is the strong genetic risk factor for IPF [7], the minor allele of the common variant in the promoter of the *MUC5B* gene (T-allele, rs35705950), which is associated with improved survival in IPF [8]. Telomeres, which cap the chromosomes and protect the genome from degradation and inappropriate recombinations/fusions, are also linked to survival [9,10]. Previously, it has been reported that a short telomere length (TL) is associated with worse survival in IPF [11,12].

Fibrotic ILDs have a more erratic clinical behavior than IPF. For instance, in fibrotic hypersensitivity pneumonitis (fHP), patients may survive up to 8–9 years after diagnosis. However, some patients deteriorate rapidly and have a disease course as seen in IPF [13,14]. Hyldgaard and colleagues have reported on survival in patients with connective tissue disease-associated interstitial lung disease (CTD-ILD). Patients with Sjögren-associated ILD had a 5-year survival rate of 84.7%, whereas patients with systemic sclerosis-associated ILD had a 5-year survival rate of 73.3% [15]. 

The recently updated ATS/ERS/JRS/ALAT clinical practice guideline presents a definition of progressive pulmonary fibrosis (PPF) in fibrotic ILD other than IPF. In short, patients having at least two of the following three criteria occurring without an alternative explanation are labelled as PPF: worsening of respiratory symptoms, lung functional deterioration and/or radiological increase in ILD/fibrosis [5]. This definition will provide clinicians with guidance in determining which patients have fast clinical deterioration and need close monitoring and/or treatment adjustment. Furthermore, it has the potential to be a prognostic tool if a correlation between PPF and outcome is established in real-world studies. 

In this study, we evaluated the proportion and survival of patients with a fibrotic ILD other than IPF who will develop PPF within the first year after diagnosis. Covariates, such as demographics, baseline characteristics, lung function, radiological and histopathological patterns, *MUC5B* (rs35705950) genotype and telomere length were investigated as potential risk factors for PPF and the results were compared with an IPF cohort.

## 2. Methods

Figure 1 shows the data collection and analysis process.

### 2.1. Patients

Screening was performed on patients diagnosed with a fibrotic ILD between January 2012 and January 2018 during a multi-disciplinary discussion at the ILD Center of Excellence (St. Antonius Hospital, Nieuwegein, The Netherlands). The included diagnoses in which the analyses were IPF, fHP, unclassifiable ILD, CTD-ILD and smoking-related ILD (SR-ILD). Patients with IPF were diagnosed according to the 2011 ATS/ERS/JRS/ALAT IPF criteria [16]. 

All patients had signs of pulmonary fibrosis on baseline high-resolution computed tomography (HRCT; ±3 months of date of diagnosis) as documented by an experienced ILD thoracic radiologist. Furthermore, all patients had provided written informed consent (approval number of the medical research ethics committee: r-05.08A).

### 2.2. Progressive Pulmonary Fibrosis

Patients were evaluated for the presence of PPF using the definition from the recently published ATS/ERS/JRS/ALAT clinical practice guideline on IPF and PPF [5]. Patients with PPF fulfilled at least 2 of the following 3 criteria during the first year after diagnosis:

(1). Worsened respiratory symptoms (cough and/or dyspnea);

(2). Either ≥ 5% absolute forced vital capacity (FVC) decline and/or ≥10% absolute diffusing capacity for carbon monoxide (DLCO) decline;

(3). Increase in fibrosis on HRCT.

We are well aware that the PPF definition is only intended for non-IPF patients with a fibrotic ILD. However, we did include an analysis of PPF in IPF for comparison purposes with the non-IPF cohort. Furthermore, diffusing capacity for carbon monoxide corrected for hemoglobin (DLCOc) was used and patients who could not perform the DLCOc maneuver due to coughing or dyspnea were noted as fulfilling the ≥10% absolute DLCO decline criterion. Furthermore, data on increase or stability in pulmonary fibrosis on HRCT were retrieved from rapports from ILD thoracic radiologists. HRCT scans performed within one year after diagnosis were compared to the baseline HRCT. In patients with clinical, lung functional or radiological progression, it was verified whether there was not an alternative explanation for the deterioration. If so, this event was not considered for the PPF analysis.

In non-PPF, follow-up time was ≥1 year for respiratory symptoms and lung function. For HRCT, we decided that a follow-up time of ≥1 year was not mandatory, because it is illogical to perform additional HRCT scans in clinically stable patients.

### 2.3. MUC5B and Telomere Length

*MUC5B* (rs35705950) T-allele carriership was determined using a pre-designed taqman single nucleotide polymorphism (SNP) genotyping assay and the QuantStudio^®^ 5 Real-Time PCR system (Thermo Fisher Scientific, Waltham, MA, USA) on DNA isolated from whole blood. Telomere length (TL; observed—expected value) was measured using a monochrome multiplex, quantitative polymerase chain reaction as described elsewhere [17]. This was performed on genomic DNA, which was extracted from peripheral white blood cells using a magnetic beads-based method (chemagic DNA blood 10k kit; Perkin Elmer Inc., Waltham, MA, USA).

### 2.4. Statistical Analysis

Baseline categorical and continuous variables were analyzed with a Chi-squared test or Fisher’s exact test where appropriate and independent Student’s t-test, respectively. Binary logistic regression analysis was used to identify covariates associated with the odds ratio (OR) for PPF. This was conducted in a non-IPF and IPF cohort. Age and percentage predicted FVC and DLCOc were stratified according to the strata used in the ILD-GAP study by Ryerson et al. [18]. The rationale behind this is that the ILD-GAP index is a commonly used prognostic tool in ILD.

Kaplan–Meier survival analysis was performed in order to obtain transplant-free (TPF) survival. Date of diagnosis was used as time point zero and lung transplantation and death were treated as events. Differences in survival between groups were determined with the log-rank test.

All analyses were conducted with SPSS version 24.0 (IBM Corp., Armonk, NY, USA) and figures were created with Graphpad Prism version 9 (Graphpad Software, San Diego, CA, USA).

## 3. Results

### 3.1. Patients

After screening (*n* = 1152), 683 patients were included in this study (see Figure 2). The non-IPF cohort is composed of 304 patients. Of them, 146 patients (46%) fulfilled the definition of PPF.

In the IPF group (*n* = 379), there were 167 patients (44%) with PPF. Baseline characteristics and clinical parameters of the non-IPF without PPF, non-IPF with PPF and IPF cohorts are provided in Table 1. Non-IPF patients without PPF had significantly higher baseline FVC and DLCOc percentages predicted compared with non-IPF patients with PPF. Furthermore, less lung transplantations and deaths occurred in the non-IPF without PPF cohort compared with the non-IPF with PPF cohort. The non-IPF with PPF and IPF comparison showed that there were significantly less males in the non-IPF with PPF cohort. Additionally, patients were younger when diagnosed with their ILD. UIP was observed in a significantly smaller proportion of patients in the non-IPF with PPF cohort than in the IPF cohort. Furthermore, the *MUC5B* minor allele frequency was also significantly lower in the non-IPF with PPF cohort compared with the IPF cohort.

Baseline characteristics of non-IPF without PPF and non-IPF with PPF were compared. Similarly, non-IPF with PPF were compared with the IPF cohort. CTD-ILD: Connective tissue disease-associated interstitial lung disease, DLCOc: Diffusing capacity for carbon monoxide corrected for hemoglobin, FHP: Fibrotic hypersensitivity pneumonitis, FVC: Forced vital capacity, HRCT: High-resolution computed tomography, IPF: Idiopathic pulmonary fibrosis, PPF: Progressive pulmonary fibrosis, SR-ILD: Smoking-related interstitial lung disease and UIP: Usual interstitial pneumonia.

### 3.2. PPF Criteria

The distribution of PPF reasons met were similar in non-IPF and IPF patients (see Figure 3). In the majority of non-IPF with PPF patients (53%), a combination of an increase in cough and/or dyspnea with lung functional decline resulted in PPF. All three criteria were met in 32% of non-IPF patients with PPF. A minority of patients fulfilled other PPF criteria. An absolute decline of ≥5% FVC and ≥10% DLCOc occurred in 54 (42%) patients, in 58 (45%) patients there was only an absolute FVC decline of ≥5% and in 18 patients (14%) there was only an absolute ≥10% DLCOc decline (see Figure 3 A,B). In IPF with PPF patients, 57% had an increase in respiratory symptoms and lung function decline within one year of follow-up. Furthermore, all three criteria were met in 28% of patients. Other PPF criteria were less frequently fulfilled. Of the 149 patients who met a lung function criterion, 52% had both ≥5% FVC and ≥10% DLCOc absolute decline and 36% solely had a ≥5% absolute FVC decline.

### 3.3. Associations with PPF Phenotype

Risk for the PPF phenotype was increased for patients with a baseline FVC < 50% compared with a baseline FVC > 75% (OR = 2.50, 95% CI = 1.01–6.17, *p* = 0.047). Furthermore, an increased odds for PPF was found for patients with a baseline DLCOc ≤ 35% (OR = 2.57, 95% CI = 1.24–5.35, *p* = 0.011) or patients who could not perform the DLCOc maneuver (OR = 2.61, 95% CI = 1.21–5.64, *p* = 0.014) compared with patients with a DLCOc >55%. There was a trend toward significance in patients diagnosed with CTD-ILD (OR = 0.60, 95% CI = 0.35–1.02, *p* = 0.06) and patients with a UIP or probable UIP on HRCT or LB (OR = 1.63, 95% CI = 1.00–2.66, *p* = 0.05; see Figure 4A). In IPF patients, baseline FVC < 50%, baseline DLCOc ≤ 35% or patients who could not perform the DLCOc maneuver were at increased odds for fulfilling criteria for the PPF phenotype (Figure 4B).

In fHP, patients who had an inciting agent from a bird had lower odds for PPF compared with the other patients with fHP (OR = 0.41, 95% CI = 0.19–0.89, *p* = 0.024, see Figure 5).

The type of underlying CTD was not a determining factor for PPF. Furthermore, patients who presented with ILD prior to their CTD-ILD did not have increased odds for PPF compared to patients who developed ILD after their CTD diagnosis (see Appendix A).

### 3.4. Survival in Non-IPF and IPF patients

In non-IPF patients, the median TPF survival was 2.9 years in patients with PPF and 10.1 years in patients without PPF (*p* < 0.001, Figure 6A). A comparison of this PPF cohort with IPF showed that survival overlaps the first three years and deviates thereafter, resulting in significantly worse TPF survival of IPF patients (*p* = 0.037, Figure 6B).

## 4. Discussion

This is the first real-world study describing survival in fibrotic ILD patients with the PPF phenotype. Patients with PPF have a median TPF survival of just 2.9 ± 0.4 years after diagnosis. Not only do PPF patients have a much worse survival than non-PPF patients, their survival in the first 3 years after diagnosis is similar to that observed in IPF patients. Later on, survival graphs diverge with IPF having slightly worse survival than PPF patients. Furthermore, we show that severely impaired lung function at diagnosis, FVC < 50% or DLCOc ≤ 35% or when not able to perform DLCOc testing, were risk factors for PPF.

Patients with fibrotic ILD other than IPF have a more variable clinical outcome than IPF patients. For fHP, it appears that patients have a disease outcome rather similar to IPF [13,14,19]. In unclassifiable ILD, an intermediate prognosis compared with IPF and other fibrotic ILD has been reported, and in CTD-ILD a relatively high five-year survival was published for multiple CTD-ILDs [15,20,21]. In this large and heterogeneous non-IPF group, PPF is helpful in identifying the patients who are at high risk for rapid clinical deterioration. 

Although not designed for IPF, we applied PPF criteria to the IPF cohort for comparison and found multiple similarities with non-IPF PPF. The proportion of patients fulfilling PPF criteria was almost identical, varying between 44 and 46%. Furthermore, the most frequently PPF criteria met in both cohorts was progressive respiratory symptoms in combination with lung function decline. Additionally, it was reported that FVC < 50% and DLCOc ≤ 35% or when not able to perform, increased the risk for PPF in non-IPF and IPF patients.

There is one obvious explanation for why impaired lung function is associated with PPF. We believe that most referred patients have end-stage pulmonary fibrosis and are progressive upon presentation in our tertiary clinic. We suspect that a proportion of patients will remain progressive during follow-up at our hospital despite treatment, due to their frailty. Furthermore, the likelihood of performing extra tests in the form of additional lung function tests or HRCT scans will be higher in these patients, which will also increase the probability to fulfill the lung function or radiological criterion of PPF.

For patients with fHP, it was reported that a causative antigen from birds decreased the odds for PPF compared to unknown or other antigens. This may be caused by a higher antigen avoidance, which can be achieved when the agent is visibly present in the patient’s environment, such as a bird. Petnak and colleagues reported that in patients with fHP, an increased risk for mortality exists when the causative antigen is not identified or not avoided by the patients [14]. In congruence with Petnak et al., our data highlight and support the importance of antigen avoidance in fHP.

The term progressive fibrosing ILD (PF-ILD), which was coined in the INBUILD trial [22], is different from PPF and emerged prior to the introduction of PPF. In the INBUILD study, it was shown that the placebo arm with a UIP-like fibrotic pattern on HRCT had similar 52-week mortality in comparison with the overall placebo group (UIP-like group: 7.8% and overall: 5.1%) [22]. Likewise, the PF-ILD study of Nasser et al. reported similar mortality after follow-up in patients with a UIP-like fibrotic pattern on HRCT or a non-UIP like pattern on HRCT (21.7%vs. 24.4%, *p* = 0.95) [23]. In accordance with these findings, we found that a UIP pattern on HRCT or LB was not a risk factor for PPF in non-IPF patients.

Furthermore, our study replicates two findings reported in two previous PF-ILD studies. First, a lower baseline lung function (FVC < 70% and DLCO < 75%) was also associated with a higher risk for PF-ILD reported in a large prospective, Canadian registry study [24]. Second, Oldham and colleagues reported that the annual FVC decline in PF-ILD patients is the greater in fHP compared with unclassifiable ILD and CTD-ILD [25]. Possibly, an unidentified trigger was also the cause of accelerated FVC decline in that study. 

Carriership of *MUC5B* (rs35705950) minor allele or decreased telomere length were not predictive for PPF. Prior studies on disease progression and survival have found similar results. In a cohort of patients with RA-ILD (*n* = 261), Juge et al. could not detect a significant association between *MUC5B* (rs35705950) genotype and annual FVC decline/survival [26]. Similarly, in a study on chronic HP (*n* = 217), a significant association between survival and *MUC5B* (rs35705950) genotype was not found [27]. In a study on patients with interstitial pneumonia with autoimmune features (IPAFs), carriership of the *MUC5B* (rs35705950) minor allele was associated with worse TPF survival, but not to changes in FVC percentage predicted per year [28], while in IPF the *MUC5B* minor allele was associated with better survival [8,29].

PPF has created the possibility for the introduction of antifibrotics in non-IPF patients. In patients with PPF, it seems a plausible treatment approach as IPF and non-IPF with PPF patients have a similar disease behavior, as shown here. Additionally, there is accumulating evidence that antifibrotics appear to be beneficial in ILD primarily driven by inflammation, fibrosis or a mixture of them [22,30,31]. Future studies should focus on the efficacy of antifibrotics in PPF, especially when combined with immunosuppression.

A couple of limitations need to be addressed. The retrospective nature of this study has resulted in missing data and patient exclusions. Selection bias may have resulted in a higher percentage of PPF in the non-IPF cohort, which was discussed above. Another limitation from this study is that sarcoidosis patients were not included in this study. A significant proportion of these patients are suspected to develop PPF [5].

In conclusion, this study is the first to report on PPF in a real-world setting. In non-IPF (*n* = 304), it was demonstrated that 46% of the patients have PPF. The transplant-free survival of these patients is significantly impaired compared with non-IPF patients without PPF (2.9vs. 10.1 years, *p* < 0.001). FVC < 50% and DLCOc ≤ 35% or patients who cannot perform the DLCOc maneuver are risk factors for PPF. Future studies are needed to validate our results and evaluate whether the identified predictors for PPF can also be applied to predict PPF in later stages of follow-up (>1 year after diagnosis).

## Figures and Tables

**Figure 1 medicina-59-00296-f001:**
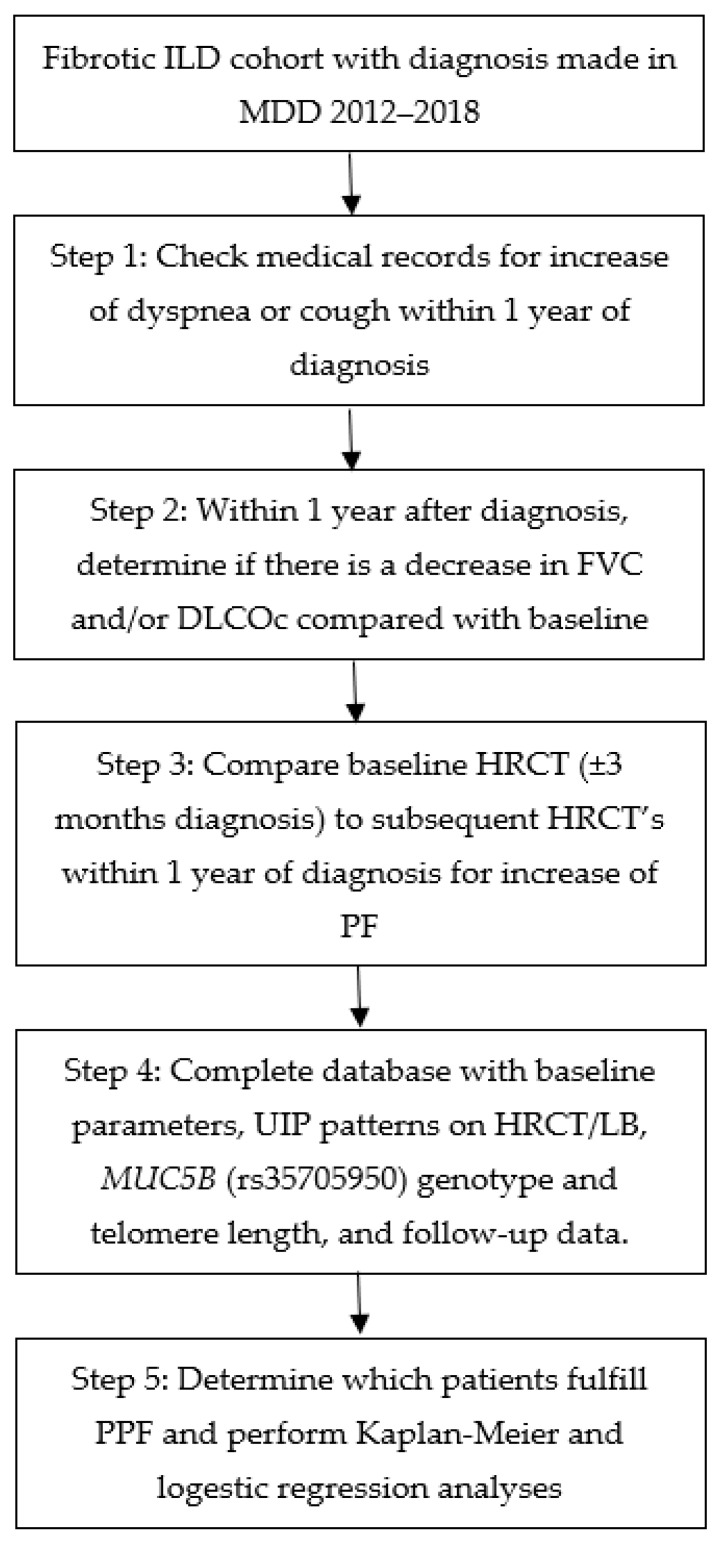
Step-by-step overview of the data collection process. DLCOc: Diffusing capacity for carbon monoxide corrected for hemoglobin, FVC: Forced vital capacity, HRCT: High-resolution computed tomography, LB: Lung biopsy, MDD: Multi-disciplinary discussion, PPF: Progressive pulmonary fibrosis and UIP: Usual interstitial pneumonia.

**Figure 2 medicina-59-00296-f002:**
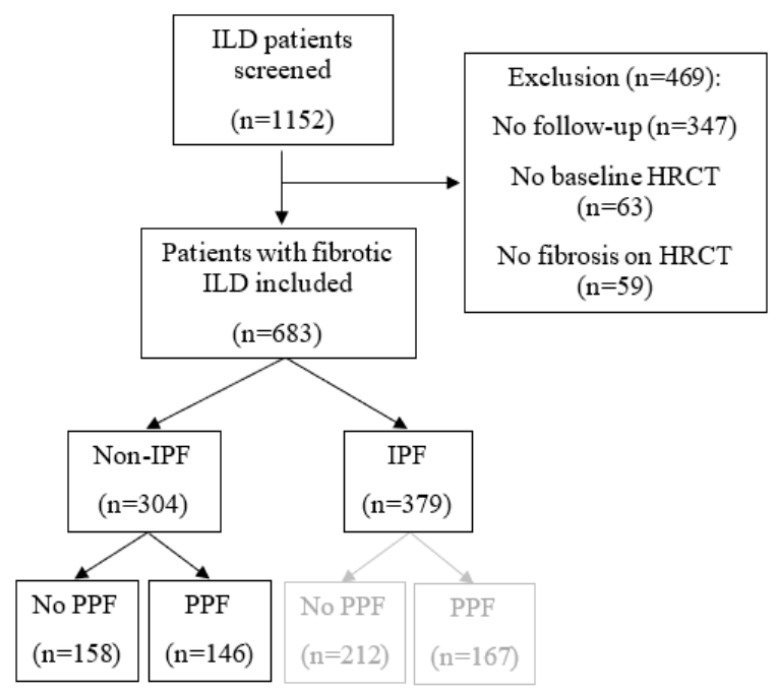
Patient flow chart. HRCT: High-resolution computed tomography, IPF: Idiopathic pulmonary fibrosis and PPF: Progressive pulmonary fibrosis.

**Figure 3 medicina-59-00296-f003:**
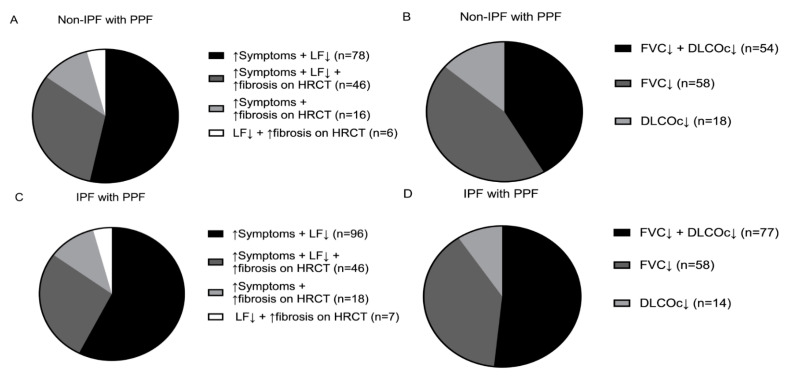
Overview of PPF and specific lung function criteria met in non-IPF and IPF. (**A**) In non-IPF, an overview of the reasons patients had PPF. Patients who experienced an increase in respiratory symptoms and lung functional decline are shown in black. Patients who fulfilled all three criteria are demonstrated in grey. In light grey, patients who had an increase in cough and/or dyspnea and fibrosis on HRCT. Patients with lung functional decline and increase of fibrosis on HRCT are illustrated with the white pie slice. (**B**) First specific PPF lung function criteria met in the non-IPF cohort with PPF. Black shows patients fulfilling both FVC and DLCOc lung function criteria. Patients who only demonstrated ≥5% absolute FVC decline are shown in grey and patients who only demonstrated ≥10% absolute DLCOc decline are shown in light grey. (**C**) The PPF criteria met in the IPF cohort. The colors represent the same criteria as mentioned under (**A**). (**D**) The specific lung function criteria met in IPF; the colors indicate the same lung function criteria stated under (**B**). DLCOc: Diffusing capacity for carbon monoxide corrected for hemoglobin, FVC: Forced vital capacity, HRCT: High-resolution computed tomography, IPF: Idiopathic pulmonary fibrosis, LF: Lung function and PPF: Progressive pulmonary fibrosis.

**Figure 4 medicina-59-00296-f004:**
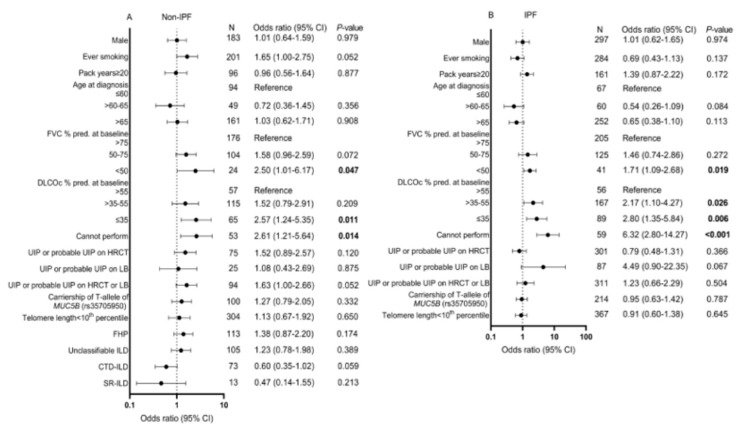
Odds ratio’s for PPF in (**A**) Patients diagnosed with a fibrotic ILD other than IPF and (**B**) Patients diagnosed with IPF. CI: Confidence interval, CTD-ILD: Connective tissue disease-associated interstitial lung disease, DLCOc: Diffusing capacity for carbon monoxide corrected for hemoglobin, FHP: Fibrotic hypersensitivity pneumonitis, FVC: Forced vital capacity, HRCT: High-resolution computed tomography, ILD: Interstitial lung disease, IPF: Idiopathic pulmonary fibrosis, LB: Lung biopsy, PPF: Progressive pulmonary fibrosis, SNP: Single nucleotide polymorphism, SR-ILD: Smoking-related interstitial lung disease and UIP: Usual interstitial pneumonia.

**Figure 5 medicina-59-00296-f005:**
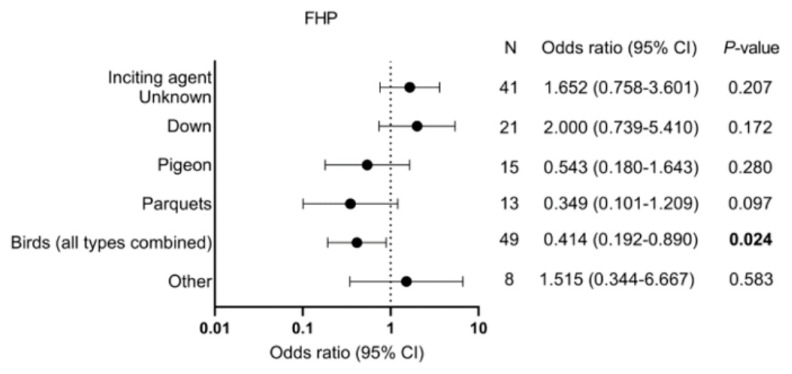
Type of inciting agents and risk for PPF. FHP: Fibrotic hypersensitivity pneumonitis and PPF: Progressive pulmonary fibrosis.

**Figure 6 medicina-59-00296-f006:**
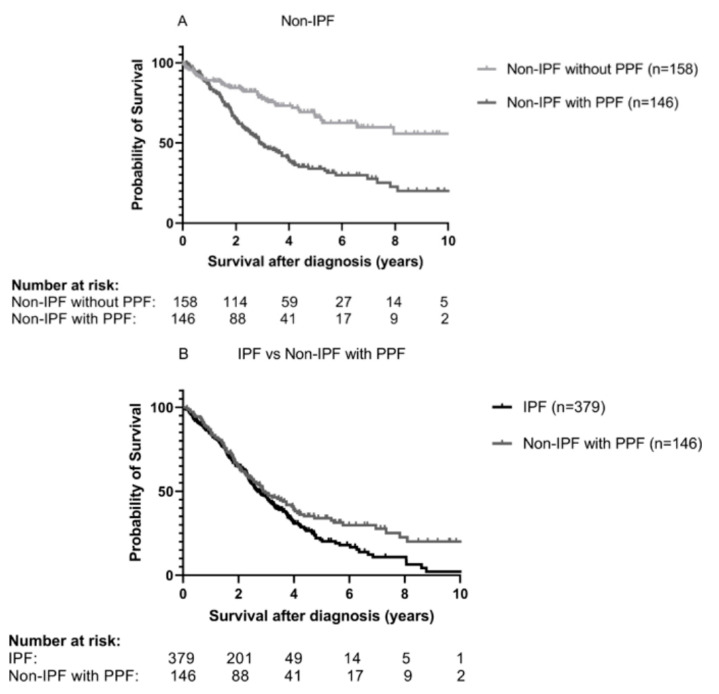
Transplant-free survival in (**A**) non-IPF, stratified into patients without PPF (light grey line, *n* = 158) and with PPF (grey line, *n* = 146). The median TPF survival was 10.1 ± 1.8 years in non-PPF and 2.9 ± 0.4 years in PPF (*p* < 0.001) and in (**B**) patients with IPF (black line, *n* = 379) and non-IPF with PPF (grey line, *n* = 146). The median TPF was 2.7 ± 0.2 years in IPF and 2.9 ± 0.4 years in non-IPF with PPF (*p* = 0.037). IPF: Idiopathic pulmonary fibrosis, PPF: Progressive pulmonary fibrosis and TPF: Transplant-free.

**Table 1 medicina-59-00296-t001:** Baseline characteristics of the study population divided into non-IPF without PPF, with PPF and IPF.

	Non-IPF without PPF (*n* = 158)	Non-IPF with PPF (*n* = 146)	*p*-Value	Non-IPF with PPF (*n* = 146)	IPF (*n* = 379)	*p*-Value
Male (%)	95 (60)	88 (60)	0.979	88 (60)	297 (78)	<0.001
Ever smokers (%)	97 (64)	104 (75)	0.051	104 (75)	284 (77)	0.612
Age at diagnosis	63.9 ± 10.1	63.8 ± 11.5	0.945	63.8 ± 11.5	68.0 ± 8.8	<0.001
Diagnosis						
FHP (%)	53 (34)	60 (41)	0.173	60 (41)	-	
Unclassifiable ILD (%)	51 (32)	54 (37)	0.388	54 (37)	-	
CTD-ILD (%)	45 (28)	28 (19)	0.058	28 (19)	-	
SR-ILD (%)	9 (6)	4 (3)	0.203	4 (3)	-	
Baseline FVC (%)	82.4 ± 22.4	75.1 ± 21.3	0.005	75.1 ± 21.3	76.2 ± 24.6	0.630
Baseline DLCOc (%)	40.5 ± 22.3	33.0 ± 20.8	0.003	33.0 ± 20.8	35.8 ± 20.0	0.153
*MUC5B* (rs35705950) genotype (minor allele = T)						
GG (%)	110 (70)	94 (64)	0.332	94 (64)	165 (44)	<0.001
GT (%)	40 (25)	49 (34)	0.115	49 (34)	181 (48)	0.003
TT (%)	8 (5)	3 (2)	0.161	3 (2)	33 (9)	0.007
Minor allele frequency	0.18	0.19	0.722	0.19	0.33	<0.001
UIP on HRCT (%)	18 (11)	26 (18)	0.112	26 (18)	215 (57)	<0.001
Underwent lung transplantation (%)	4 (3)	15 (10)	0.005	15 (10)	21 (6)	0.055
Deceased (%)	44 (28)	80 (55)	<0.001	80 (55)	217 (57)	0.610

## Data Availability

Data sharing is not applicable to this article.

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
