# Peer review of "Decreased Survival and Lung Function in Progressive Pulmonary Fibrosis"

_medicina, 2023, doi:10.3390/medicina59020296_

Round 1
Reviewer 1 Report
Type of manuscript: Article
Title: Decreased survival and lung function in progressive pulmonary fibrosis
Journal: Medicina
As for the paper, its subject is important and requires a great epidemiological work to obtain the information.
However, it could be improved:
A) As for the conclusion of the abstract does not necessarily reflect the objectives set in the work.
B) At the end of the discussion it does not close with clear conclusions that reflect the objectives set.
C) Although the methodology is adequate, perhaps a scheme could be made where the most interesting results obtained are more clearly indicated.
Author Response
Dear reviewer 1,
Thank you for your kinds words. Hereby a point by point response to the comments.
A: We revised the conclusion focusing more on the survival difference and prognostics biomarkers found for PPF.
B: Similar to the abstract, we now clearly state the survival difference and which biomarkers we found.
C: We added figure 1, which is a schematic overview of the data collection and analysis process.
Reviewer 2 Report
The paper is interesting and the definition of prognostic biomarkers represents a very important key point in the diagnosis and the definition of the therapeutic perspective.
It would be appreciated a consideration (to integrate into the discussion) by the authors on a perspective of potential differences in pharmacological approach, that takes into account the role of PPF in pathology progression for patients with fibrotic ILD other than idiopathic pulmonary fibrosis.
Author Response
Dear reviewer 2,
Thank you for your kind words. We added a paragraph in the discussion about antifibrotics in PPF.